# Protective Effects of Fucoxanthin Dampen Pathogen-Associated Molecular Pattern (PAMP) Lipopolysaccharide-Induced Inflammatory Action and Elevated Intraocular Pressure by Activating Nrf2 Signaling and Generating Reactive Oxygen Species

**DOI:** 10.3390/antiox10071092

**Published:** 2021-07-07

**Authors:** Shiu-Jau Chen, Tzer-Bin Lin, Hsien-Yu Peng, Cheng-Hsien Lin, An-Sheng Lee, Hsiang-Jui Liu, Chun-Chieh Li, Kuang-Wen Tseng

**Affiliations:** 1Department of Neurosurgery, Mackay Memorial Hospital, Taipei 10449, Taiwan; chenshiujau@gmail.com; 2Department of Medicine, Mackay Medical College, New Taipei 25245, Taiwan; hypeng@mmc.edu.tw (H.-Y.P.); davidlin@mmc.edu.tw (C.-H.L.); anshenglee@mmc.edu.tw (A.-S.L.); 1105150343@live.mmc.edu.tw (C.-C.L.); 3Department of Physiology, School of Medicine, College of Medicine, Taipei Medical University, Taipei 11049, Taiwan; tblin2@tmu.edu.tw; 4Department of Optometry, MacKay Junior College of Medicine, Nursing, and Management, New Taipei 11260, Taiwan; s458@eip.mkc.edu.tw

**Keywords:** fucoxanthin, lipopolysaccharide, uveitis, oxidative stress, nuclear factor erythroid 2-related factor 2

## Abstract

Inflammation and oxidative stress are closely related processes in the pathogenesis of various ocular diseases. Uveitis is a disorder of the uvea and ocular tissues that causes extreme pain, decreases visual acuity, and can eventually lead to blindness. The pharmacological functions of fucoxanthin, isolated from brown algae, induce a variety of therapeutic effects such as oxidative stress reduction and repression of inflammation reactions. However, the specific anti-inflammatory effects of fucoxanthin on pathogen-associated molecular pattern (PAMP) lipopolysaccharide-induced uveitis have yet to be extensively described. Therefore, the aim of present study was to investigate the anti-inflammatory effects of fucoxanthin on uveitis in rats. The results showed that fucoxanthin effectively enhanced the expression of nuclear factor erythroid 2-related factor 2 (Nrf2) in ocular tissues. Furthermore, fucoxanthin significantly increased the ocular activities of superoxide dismutase and decreased the levels of malondialdehyde stimulated by PAMP-induced uveitis. Ocular hypertension and the levels of inflammatory cells and proinflammatory cytokine tumor necrosis factor-alpha in the aqueous humor were alleviated with fucoxanthin treatment. Consequently, compared to the observed effects in lipopolysaccharide groups, fucoxanthin treatment significantly preserved iris sphincter innervation and pupillary function. Additionally, PAMP-induced corneal endothelial disruption was significantly inhibited by fucoxanthin treatment. Overall, these findings suggest that fucoxanthin may protect against inflammation from PAMP-induced uveitis by promoting the Nrf2 pathway and inhibiting oxidative stress.

## 1. Introduction

Inflammation, a key factor in many diseases, may be accompanied by oxidative stress; together, these processes act as cooperative and even synergistic partners in worsening the pathogenesis of several diseases. Lipopolysaccharides (LPSs), i.e., heat-stable cell wall components of Gram-negative bacteria, which are known as endotoxins and pathogen-associated molecular patterns (PAMPs), are often employed in the search to induce a variety of pathophysiological effects, inflammatory diseases, and autoimmune disorders [1,2,3,4]. The pathogenesis of LPS-induced injury can include oxidative stress and inflammatory responses. Furthermore, exacerbated production of reactive oxygen species (ROS), which play essential roles in the regulation of immune responses against pathogens under physiological conditions and the progression of inflammatory disorders, can lead to protein and lipid disruption, proinflammatory cytokine expression, inflammatory infiltration, physiological process dysfunction, organelle damage, and cell death [5,6,7].

Uveitis is a general term describing a group of intraocular inflammatory diseases that often affect the uvea [8]. Injection of LPS can generate PAMP-induced uveitis, which is widely used as an animal model to examine the pathological mechanisms of uveitis and ophthalmic inflammation in animals [9,10,11]. In a state of inflammation, the uveal tract, including the iris, shows the accumulation of inflammatory cells. Iris atrophy and sphincter muscle paralysis are often observed in patients with uveitis [12]. Additionally, an increase in the number of trabecular precipitates (which are composed of various proteins, inflammatory cells, and fragments) in the anterior chamber can result in decreased trabecular outflow due to blockage of the trabecular meshwork [13,14]. Obstruction by inflammatory precipitates associated with iritis is often accompanied by high intraocular pressure (IOP). Moreover, corneal endothelial cell junction disruption and corneal edema are also observed during uveitis [15,16]. In young- to middle-aged patients, uveitis can cause extreme pain and light sensitivity that leads to the loss of the patient’s social and economic independence because of partial-to-total blindness [17]. Thus, the identification of novel treatments for uveitis is a priority.

One such potential therapeutic strategy involves targeting endogenous ROS to stabilize the microenvironment and thereby ameliorate pathophysiological effects. Among various antioxidative pathways, the nuclear factor erythroid 2-related factor 2 (Nrf2) pathway includes a key cytoprotective transcription factor that functions in the amelioration of various oxidative stress- and inflammation-associated diseases [18,19]. Once released, Nrf2 moves into the cell nucleus and binds to the DNA at the location of the antioxidant response elements to enhance the expression of antioxidant enzymes, including superoxide dismutase (SOD) and glutathione peroxidase, during oxidative stress. Numerous studies have demonstrated that Nrf2 can enhance the inhibition of the inflammatory response in several tissue types in ocular disease, including photokeratitis, cataracts, and retinopathy [20,21,22,23]. Nrf2 meditates the cellular signaling pathways that significantly reduce cell loss, inhibit proteolysis, and improve permeability barrier integrity in various ocular tissues.

Fucoxanthin, an orange-colored pigment, is the most abundant marine carotenoid; it is responsible for the high antioxidant properties of brown algae [24]. Fucoxanthin has noteworthy biological characteristics based on its unique molecular structure; it contains an unusual allenic bond and a 5,6-monoepoxide structure, which differs from that of other carotenoids. Our previous studies have demonstrated that fucoxanthin pretreatment impedes ultraviolet B (UVB)-induced corneal inflammatory pain and exfoliation of the corneal epithelial layer [22,25]. Moreover, fucoxanthin can induce a variety of therapeutic effects such as reducing oxidative stress, repressing inflammation reactions, and protecting the digestive tract as well as the blood vessels of the neural, skeletal, and integumentary system. The effects of fucoxanthin on LPS-induced leucocyte and protein infiltration in rats’ aqueous humor have been studied [26,27]. However, the specific effects of fucoxanthin on uveitis have yet to be examined in detail. Therefore, in this study, we investigated whether fucoxanthin has anti-inflammatory effects on PAMP-induced uveitis and attempted to elucidate the anti-inflammatory mechanism of such effects.

## 2. Materials and Methods

### 2.1. Experimental Animals

All experimental animals were cared for and treated in accordance with the recommendations in the Guide for the Care and Use of Laboratory Animals of the National Institutes of Health. The animal experiment was approved by the Institutional Animal Care and Use Committee of Mackay Medical College (New Taipei City, Taiwan; permit number: IACUC-A1080019). Healthy male Sprague Dawley rats (aged 4–6 weeks; body weight (BW), 180–200 g) were purchased from the animal department of BioLASCO Taiwan Co., Ltd. (Taipei, Taiwan), after which they were quarantined and then allowed to acclimatize for 5 days before experimentation began. Experimental animals were allocated at 3–4 per cage, and maintained at 19–23 °C (room temperature), 40–50% relative humidity, with a 12 h light and 12 h dark cycle, and with ad libitum access to drinking water and food.

### 2.2. Induction of Endotoxin-Induced Ocular Disorders and Experimental Design

Twenty-five experimental animals were randomly allocated to one of five groups: A blank control group (without LPS injection or fucoxanthin treatment), an LPS/vehicle group (LPS injection and pretreatment by gavage of 0.1 mL of physiological phosphate-buffered saline (PBS)/day for 2 days prior to the experiment), or LPS/fucoxanthin groups (LPS injection and pretreatment by gavage of fucoxanthin (Sigma-Aldrich, St Louis, MO, USA) at 0.1, 1, or 10 mg/kg BW in a 0.1% dimethyl sulfoxide solution (Sigma-Aldrich) mixed with 0.1 mL of PBS/day for an interval of 2 days prior to experimentation). The dose of fucoxanthin was chosen based on our previous study of ocular diseases in a rat model [22]. After anesthesia with an intraperitoneal injection of sodium pentobarbital (50 mg/kg BW), PAMP LPS-induced uveitis was induced with a subcutaneous injection into the footpad of 200 μg of LPS (100 μg/footpad) obtained from *Escherichia coli* (Sigma-Aldrich) and diluted in 200 μL of sterile pyrogen-free saline. Experimental animals were euthanized and assayed 5 days after the induction of uveitis.

### 2.3. Measurement of Nrf2 Protein Levels

Both cytoplasmic and nuclear extracts were prepared in parallel using extraction reagent kits (Thermo Fisher Scientific, Rockford, lL, USA) according to the manufacturer’s protocol. An enzyme-linked immunosorbent assay (ELISA) was used to measure Nrf2 protein levels in the supernatant of cytoplasmic and nuclear extracts from experimental tissues following the manufacturer’s instructions (Novous, Centennial, CO, USA). Optical density was measured spectrophotometrically at a wavelength of 450 nm with a microplate reader. The sensitivity of the assay for Nrf2 was 9.3 pg/mL.

### 2.4. Measurement of IOP

The IOP at the center of the cornea was measured using a Tonolab rebound tonometer (Icare, Helsinki, Finland) in accordance with the manufacturer’s recommendations. While the rats were awake, 15 μL of 0.5% proparacaine hydrochloride was topically applied to each ocular surface prior to taking IOP measurements. To achieve IOP measurements while the rats were awake, each animal was restrained by holding the skin between its neck and tail, while avoiding compression of the neck and the thoracic and abdominal cavities. Each measurement was repeated three to four times and then averaged. To control for any diurnal discrepancy in the IOP, all measurements were recorded between 11:00 and 12:00.

### 2.5. Determination of SOD and Oxidative Stress-Related Malondialdehyde (MDA) 

According to the weight of the anterior segment, the activities of SOD and the levels of MDA in ocular tissues were measured using commercialized assay kits according to the manufacturer’s instructions (Sigma-Aldrich). Briefly, the tissues were homogenized in ice-cold 0.1-M Tris/HCl (pH 7.4, containing 5 mM of β-mercaptoethanol, 0.1 mg/mL of phenylmethylsulfonyl fluoride, and 0.5% Triton X-100). After centrifugation at 4 °C (14,000× *g* for 5 min), the supernatant and SOD reagents were measured by determining the absorbance at 450 nm. A portion of the homogenate was immediately measured for its MDA levels. The supernatants were supplemented with thiobarbituric acid and then boiled at 95 °C in a water bath for 60 min. The reaction of MDA with thiobarbituric acid was measured by determining the absorbance at 532 nm.

### 2.6. Inflammatory Cell Counts in the Aqueous Humor

Cell counting in the aqueous humor was performed as previously described with slight modifications [28]. The experimental animals were anesthetized with an overdose of choral hydrate (450 mg/kg BW) via intraperitoneal injection, after which they were sacrificed by cervical dislocation. Immediately after sacrifice, the aqueous humor was acquired by penetrating the cavity between the cornea and the iris with a 30 gage needle. For cell counting, 1 µL of aqueous humor was diluted with 9 µL of PBS and then suspended in 10 µL of trypan-blue solution (Sigma-Aldrich). Subsequently, the cell suspension was assessed with a hemocytometer. The number of cells per square was quantified manually under a light microscope and the total number of cells in five squares per sample was averaged.

### 2.7. Immunohistochemistry of Myeloperoxidase (MPO) and Zonula Occludens-1 (ZO-1) in the Ocular Tissues

The experimental rats were deeply anesthetized via an intraperitoneal injection of choral hydrate (400 mg/kg BW) and perfused with a fixative containing 4% paraformaldehyde in PBS. Tissue sections were pretreated with 3% H_2_O_2_ for 10 min at room temperature to exhaust endogenous peroxidase activities. After incubation in PBS containing 5% skim milk at 37 °C for 30 min, the sections were treated with primary antibody against tissue leukocyte marker MPO (Abcam, Cambridge, U.K.) or intercellular junction ZO-1 (Abcam) for 2 h at room temperature, followed by 3 washes in PBS. The sections were then incubated with horseradish peroxidase-conjugated goat secondary antibody for 1 h at room temperature. After being rinsed in PBS, the samples were placed in 0.05% DAB/0.01% H_2_O_2_ solution for color development.

To evaluate the integrity of the corneal endothelium, corneal whole-mount samples were washed and incubated with intercellular junction ZO-1. DAPI (40,6-diamidino-2-phenylindole, Thermo Fisher Scientific, Waltham, MA, USA) was used to stain nucleic acids for the nuclear staining.

### 2.8. Pupillometry of the Pupillary Light Reflex

Pupillary reactions were evaluated in unanesthetized rats following our previous protocol [29] with some modifications. Each rat was adapted to darkness for at least 45 min and subsequently placed on a custom-built stereotactic apparatus while their motion was confined using a 56 mm-diameter polyethylene tube. The eye was gently held open during monitoring. A beam of light was directed to the center of the ocular surface for evaluation of the pupillary response. All data were recorded from the right pupil with the eye positioned at an equivalent distance from the digital camera. The pupillary diameter was measured and used to calculate the pupil area.

### 2.9. Immunohistochemical Analysis of the Iris Nerve Fibers

To determine the extent of innervation, the iris tissues were collected for immunohistochemical analysis as described in our previous report [29]. Iridial whole-mount samples were transferred into PBS solution containing 3% hydrogen peroxide to eliminate endogenous peroxidase activity. After blocking nonspecific binding using 10% bovine serum albumin (Sigma-Aldrich) and 1% Triton X-100 in PBS, the samples were washed and incubated with a specific primary antibody against pan neuronal marker proteins, protein gene product (PGP) 9.5 (dilution, 1:250; Chemicon International, Inc., Temecula, CA, USA), at 4 °C for 2 days. After washing with PBS, the samples were incubated with an appropriate biotinylated secondary antibody (Sigma-Aldrich). The color reaction products were visualized with 3,3′-diaminobenzadine using a VECTASTAIN^®^ ABC Kit (Vector Laboratories, Burlingame, CA, USA). A flat mount of iris tissues was evaluated under a Zeiss Axiophot microscope (Carl Zeiss, Oberkochen, Germany). The total lengths of nerves labeled with antibody against PGP 9.5 were calculated with commercial digital software (Adobe Illustrator; Adobe Systems, San Jose, CA, USA) using the object–length function, as described in a previous protocol [30] but with minor modifications. The innervation density of the iris sphincter was then recorded as a percentage relative to the control.

### 2.10. Statistical Analysis

All data are expressed as means ± standard deviation (SD). A Kolmogorov–Smirnov test was used to verify the normality of the data. Nonparametric values were analyzed with a Mann–Whitney test. In contrast, parametric values were analyzed with Student’s *t*-test or one-way ANOVA followed by Bonferroni’s multiple comparison test. Differences were considered statistically significant at *p* < 0.05. SPSS (SPSS, Inc., Chicago, IL, USA) was used to perform all statistical analyses.

## 3. Results

### 3.1. Fucoxanthin Upregulates the Expression of Nrf2 and Enhances the Nuclear Translocation of Nrf2 in PAMP LPS-Induced Uveitis

As previously mentioned, PAMP LPS-induced uveitis is a classical model for the study of noninfectious ocular inflammation. In the present study, we investigated whether fucoxanthin induces the expression of Nrf2, which is an essential mediator of cellular reactions against oxidative stress and inflammatory responses. We measured the relative expression levels of cytoplasmic Nrf2 and nucleic Nrf2 with an ELISA assy. The results showed that fucoxanthin pretreatment promotes activation of cytosolic Nrf2 (Figure 1A) and enhances the nuclear translocation of Nrf2 (Figure 1B).

### 3.2. Protective Effects of Fucoxanthin on LPS-Induced Elevated IOP

The mean IOP values of the rats in the blank control group were within the normal range (14.6 ± 1.2 mmHg), but these values were significantly higher among the rats in the PAMP LPS-induced uveitis group (23.2 ± 1.3 mmHg). Although LPS injection caused a significant increase in IOP, the effects of LPS were ameliorated by fucoxanthin. The IOP remained elevated in the groups treated with 0.1 and 1 mg/kg BW of fucoxanthin (21.3 ± 1.5 and 20.2 ± 1.1 mmHg, respectively) after LPS treatment, but the IOP was significantly reduced in the groups treated with 10 mg/kg BW of fucoxanthin (15.5 ± 0.8 mmHg) (Figure 2).

### 3.3. Protective Effects of Fucoxanthin on SOD Activity and MDA Levels

LPS was found to induce an increase in IOP that could be inhibited by fucoxanthin. Thus, we also examined the effects of fucoxanthin on antioxidative capabilities by measuring the levels of SOD and MDA in treated rats, because, in the ocular tissues, these levels are indicative of antioxidative and oxidative damage capacities, respectively. The SOD activity in the eyes of the LPS/vehicle group rats was notably lower than that in the control group (*p* < 0.05); however, the group pretreated with 10 mg/kg BW of fucoxanthin showed a remarkable increase in SOD activity relative to the activity measured in the LPS/vehicle group (Figure 3A). In addition, the levels of MDA in the ocular tissue of the LPS/vehicle group were significantly increased compared to the MDA levels in the blank control group, but these levels were significantly decreased (*p* < 0.05) in the ocular tissue of the rats pretreated with 10 mg/kg BW of fucoxanthin (Figure 3B).

### 3.4. Histological Analysis of Ocular Tissues, Inflammatory Cell Counts, and Proinflammatory Cytokine Tumor Necrosis Factor-alpha (TNF-α) Protein Concentration in the Aqueous Humor

Compared to the blank control group (Figure 4A), iris hyperemia was observed in the LPS group (Figure 4B); however, this effect was alleviated in the 10 mg/kg BW of fucoxanthin-treated group (Figure 4C). To characterize the morphological changes that occur in uveitis, the anterior ocular structures of transverse sections were examined. Measurements of the histological examinations demonstrated that after treatment with LPS, significant differences in angle closure and increased MPO-positive inflammatory cells (Figure 4E,H) existed in comparison to these measurements in the control group (Figure 4D,G). In addition, decreases in the angle closure and anterior chamber depth, as well as an increase in corneal thickness, were observed in the LPS-induced autoimmune iritis group. However, when treated with fucoxanthin, a remarkable reduction in angle closure (Figure 4F) and inhibition of infiltrating cells in the aqueous humor and iris regions were observed (Figure 4F,I), relative to these measurements in the LPS/vehicle experimental groups.

To confirm the infiltration of anti-inflammatory cells and protein concentrations of ocular tissues in PAMP LPS-induced ocular disorders, the number of infiltrating cells and the expression levels of TNF-α were measured in the aqueous humor. After LPS injection, the number of infiltrating cells increased in the aqueous humor of the LPS/vehicle group (14.7 ± 3.2 × 10^5^ cells/mL; *n* = 5), but significantly decreased (relative to the LPS/vehicle group) in the aqueous humor of the fucoxanthin-treated groups (2.1 ± 1.6 × 10^5^ cells/mL; *n* = 5; *p* < 0.05) (Figure 4J). In addition, TNF-α was strongly expressed in the LPS/vehicle group. In contrast, fucoxanthin treatment significantly reduced the concentrations of inflammatory cytokines in the aqueous humor (Figure 4K). Collectively, the histopathological and ELISA assay findings indicate that fucoxanthin mitigates ocular inflammation.

### 3.5. Effect of Fucoxanthin on the PAMP LPS-Induced Impaired Pupillary Light Reflex and Autonomic Denervation of Iridial Tissues

This pupillary light test can be used to diagnose parasympathetic denervation of the pupillary sphincter muscle. To assess denervation in the area of the sphincter muscles of the iris, the initial noticeable symptom of autonomic impairment is a reduction in the amplitude of pupillary constriction to light. In the present study, the pupillary light reflex was affected by LPS treatment. After LPS treatment, the harshness of pupil abnormalities and other autonomic deficits increased (Figure 5C,D). Compared to the control (Figure 5A,B), the initial pupil diameter was significantly larger, the constriction velocity was decreased, and the peak constriction amplitude to light stimuli was diminished in the LPS/vehicle group. In contrast, the impaired reflex was alleviated with fucoxanthin treatment (Figure 5E,F). Additionally, the pupil diameter of pupillary light differed significantly between the LPS/fucoxanthin group and the LPS/vehicle group (Figure 5G). These results indicate that LPS-induced impairment of the pupillary light reflex was suppressed by fucoxanthin.

To define the effect of fucoxanthin on LPS-induced degeneration of the autonomic nerve, the general neuronal marker PGP 9.5 was used as a target for histopathological analysis. In the control group, there was an abundance of autonomic nerves in the sphincter muscles of the iris; staining of PGP 9.5 was dense with continuous patterns (Figure 5H). In the whole-mount iris of the LPS/vehicle group, there was a noticeable decrease in the density of immunopositive fibers throughout the sphincter region compared to observations in intact control irises (Figure 5I). In the fucoxanthin treatment group, denervation was significantly reduced and nerve density increased in the sphincter area (Figure 5J) relative to these measurements in the LPS/vehicle group.

Compared to the control group, the decrease in autonomic nerve density was statistically significant in the LPS/vehicle group (23.4% ± 4.6%; *p* < 0.05). In contrast, compared to the LPS group, denervation in the sphincter region of the iris was reduced in the 10 mg/kg BW of fucoxanthin pretreatment group (75.1% ± 12.8%; *p* < 0.05) (Figure 5K). These findings suggest that LPS-induced denervation in the sphincter region of the iris was efficiently inhibited with fucoxanthin treatment.

### 3.6. Effects of Fucoxanthin on the Cell Infiltration and Endothelial Cell Junctions of Corneal Tissues

To examine the effect of fucoxanthin on LPS-induced cell infiltration of corneal tissues, inflammatory cells were quantified by immunohistochemical analysis. In the blank control group, no evidence of MPO-positive leukocytes was detected (Figure 6A), but a continuous labeling of ZO-1 around corneal endothelial cells was observed (Figure 6D,G). The results showed a marked increase in the number of MPO-positive cells induced by LPS treatment (Figure 6B) compared to the number detected in the blank control group. The corneal stroma was strongly infiltrated by MPO-positive leukocytes as a result of LPS treatment. The abundance of infiltrated MPO-positive leukocytes was significantly increased in the corneal stroma; this was also correlated with the intensity of endothelial junction disruption, because cell junction ZO-1 was lost in the LPS/vehicle group (Figure 6E,H). Compared to the LPS/vehicle group, treatment with fucoxanthin resulted in significantly lower numbers of adherent and infiltrated leukocytes, as well as enhanced disruption of the endothelial junctions in the corneas (Figure 6C,F,I). Fucoxanthin treatment, therefore, decreased the number of inflammatory cells and increased the intensity of cell–cell junction ZO-1 expression.

## 4. Discussion

In general, the basal expression of antioxidant elements is not appreciably regulated by Nrf2 signaling [31]. However, Nrf2 is involved in regulating the innate immune response, and ROS have been shown to accumulate at higher rates in the retina and ciliary body of Nrf2-deficient mice [32]. Nrf2–Keap1 is considered to be one of the most critical transduction pathways in regulating the oxidative stress response of cells. A recent study found that fucoxanthin specifically targets Keap1 and inhibits the interaction between Keap1 and Nrf2 [33]. Nrf2 and its interaction with anti-oxidant response elements increase the transcription of the phase II antioxidant enzymes, such as SOD, NAD(P)H, NAD(P)H: quinone oxidoreductase 1, and heme oxygenase-1 [34]. A previous study conducted by our group demonstrated that pretreatment with fucoxanthin effectively protects the corneas from denervation and inhibits trigeminal pain in UVB-induced keratitis with significant increases in Nrf2 expression in vivo [22]. Building on this work, in the present study, we investigated the anti-inflammatory effects of fucoxanthin against PAMP-induced ocular disorders, as well as the possible underlying mechanisms, in vivo. The results of our study demonstrated that fucoxanthin substantially ameliorates noninfectious inflammatory responses in experimental animals treated with LPS, with significant reductions in oxidative stress, inflammatory cell infiltration, and protein TNF-α concentrations observed in the aqueous humor. In addition, impairment of the pupillary light reflex and denervation of the autoimmune nerves of iris sphincter muscles were reduced by fucoxanthin. However, the functional changes are not completely dependent on a fucoxathin-mediated increase in Nrf2 expression and elevated SOD activity. It has been shown that fucoxanthin also activates the AMP-activated protein kinase signaling pathway and inhibition of mitochondria dysfunction induced by oxidative stress [35]. Further studies will be crucial to clarify the effects of fucoxanthin on the expression of other genes involved in the anti-inflammatory effects.

Inflammation plays a critical role in the innate immune response by preventing tissue destruction and promoting tissue healing [36]. The goals of therapy for autoimmune uveitis are to reduce inflammation, prevent impairment of ocular tissues, inhibit elevations in the IOP, and prevent long-term visual loss. To date, various corticosteroid regimens have been the mainstay for uveitis treatment; however, the uncontrollable side effects of these corticosteroid treatments can give rise to advanced tissue damage. The most frequent ocular side effects are glaucoma, cataracts, poor wound healing, ptosis, and mydriasis [37,38]. In addition, a break in the functional integrity of the corneal endothelium is associated with corneal edema [39]. Corneal thickness plays a critical role in the anterior chamber angle and the rate aqueous humor flow out of the eye. In the present study, the risk of glaucoma development among the experimental animals with autoimmune uveitis, as well as a narrowed anterior chamber angle, elevated IOP, and ocular inflammatory responses, were significantly reduced by treatment with fucoxanthin. 

LPS from Gram-negative bacteria is an important stimulus for tissue inflammation. Higher doses of LPS induce significant systemic disorders involving the brain, liver, and testes [40,41,42]. However, footpad injection of LPS at a relatively low dose is appropriate for inducing ocular inflammation with no other significant systemic disorders [43]. Oxidative stress and proinflammatory cytokines have been implicated in the development of ocular inflammation [44,45]. The proinflammatory cytokine TNF-α in the aqueous humor plays an important role in the pathogenesis of anterior uveitis. TNF-α levels are significantly elevated in the aqueous humor in the early stages of endotoxin-induced anterior uveitis [46,47]. Furthermore, the concentration of proinflammatory cytokines in the aqueous humor has been shown to increase before apparent clinical signs of anterior uveitis emerge; moreover, blocking signaling results in a significant delay in the onset of anterior uveitis pathologies [48,49]. Other studies have also assessed the effects of TNF-α on vascular leakage, leukocyte adhesion, and the recruitment of circulating inflammatory factors in ocular tissues [50,51]. In our own prior studies, we demonstrated that fucoxanthin suppresses UVB-induced TNF-α expression and the deterioration of epithelial smoothness in the photokeratitis of corneal tissues [25]. In the present study, antioxidative capabilities were determined by measuring the activity of SOD and the levels MDA in ocular tissues. Both the expression levels of TNF-α and the number of inflammatory cells decreased in the aqueous humor following fucoxanthin treatment, suggesting that fucoxanthin regulates oxidative stress and inflammatory cytokines while inhibiting the recruitment of leukocytes in a rat model of endotoxin-induced ocular disorder.

Anterior uveitis is caused by several potential etiologies and may involve other adjacent tissues. Examination of the iris preceding dilation can sometimes facilitate an etiologic diagnosis. Most types of acute anterior uveitis are caused during transient pupillary miosis. Anterior uveitis associated with sectoral iris atrophy is frequently ascribed to antibiotic- and virus-induced uveitis. Disruptions of collagen tissue in the smooth muscle and the blood vessels of the sphincter pupillae are known causes of iris atrophy and sphincter muscle paralysis in patients with moxifloxacin-induced uveitis [12]. Histological analysis via microscopic examination has revealed destruction of the iris muscle and necrosis of the iris sphincter in clinical uveitis [52]. Moreover, dysfunction of the ocular autonomic nerves that regulate the pupillary reflex results in abnormal pupil size [53]. In the present study, in the LPS-induced uveitis group, nerve innervation and the reduction in autonomic nerve density were significantly reduced in the area of the sphincter muscles of the iris and pupil size was abnormal in response to light stimuli. However, LPS-induced denervation and impairment of the pupillary light reflex were suppressed in the fucoxanthin-treated group.

Such negative pressure can induce fluid leakage into the stroma from the anterior chamber, but the rate of leakage is restrained by the tight junctions of the endothelium. The corneal endothelium is localized to the inner surface of the cornea into the anterior chamber of the eye and is an immediate target of ophthalmic inflammation during anterior uveitis. Close associations exist between the clinical signs of corneal disease and the onset of uveitis. The corneal endothelium is susceptible to cellular damage due to chronic inflammation, which is further supported by the detection of immune imbalance. In a previous study, it was demonstrated that cellular adhesion molecules, such as E-selectin, are expressed in the corneal endothelium after endotoxin injection [54]. Consistently, the formation of keratic precipitates in the corneal endothelium is characteristic of uveitis [16]. Anterior uveitis involves infiltration of polymorphonuclear inflammatory cells, increased protein permeability, and upregulation of inflammatory cytokines in the aqueous humor [55,56]. In the present study, disrupted endothelial tight junction ZO-1, infiltrated inflammatory cells, and edema of the cornea were significantly induced by LPS treatment; however, these effects were suppressed by fucoxanthin treatment. Dispersion of ZO-1 and swelling of cornea in response to TNF-α involves activation of matrix metalloproteinases-9 and p38 mitogen-activated protein kinase [57,58]. Herein, we discovered that fucoxanthin protects corneal barrier function perhaps based on fucoxanthin-mediated control of the TNF-α inflammatory cytokine concentrations in the aqueous humor.

## 5. Conclusions

The results of the present work show that treatment with fucoxanthin prevents PAMP-induced uveitis. A significantly decreased number of inflammatory cells and an improved narrow anterior chamber angle were observed in the groups treated with fucoxanthin following the development of ocular disorders induced by LPS. Moreover, iris innervation and the pupillary light reflex were preserved in the fucoxanthin-treated groups. These findings suggest that fucoxanthin may protect the eyes from PMAP-induced inflammatory action, as well as an elevated IOP.

## Figures and Tables

**Figure 1 antioxidants-10-01092-f001:**
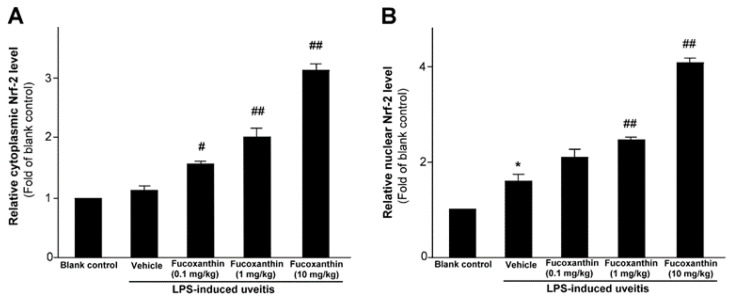
Effect of fucoxanthin on Nrf-2. Cytosolic (**A**) or nuclear proteins (**B**) were determined by ELISA and are expressed as the relative level of the controls. The experiments were repeated three times and similar results were obtained. The results are presented as the mean ± SD (*n* = 5). * *p* < 0.05: Significant difference compared to the blank control group (Student’s *t*-test); # *p* < 0.05 and ## *p* < 0.01: Significant difference in the LPS-treated groups (one-way ANOVA followed by Bonferroni’s multiple comparison test).

**Figure 2 antioxidants-10-01092-f002:**
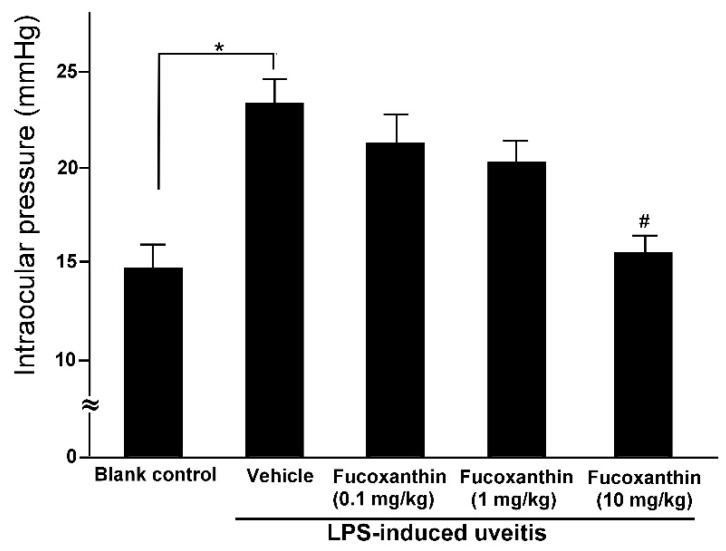
Effects of fucoxanthin on IOP after LPS-induced uveitis. The IOP values were compared among the blank control, LPS/vehicle, LPS/0.1 mg/kg BW of fucoxanthin, LPS/1 mg/kg BW of fucoxanthin, and LPS/10 mg/kg BW of fucoxanthin groups. Compared to the blank control group, the IOP was significantly elevated in the LPS-treated group, but the IOP then decreased with fucoxanthin treatment. The results are presented as the means ± SD (*n* = 5 rats/group). * *p* < 0.05: Significantly different compared to the blank control group (Student’s *t*-test). # *p* < 0.05: Significantly different in the LPS-treated groups (one-way ANOVA followed by Bonferroni’s multiple comparison test).

**Figure 3 antioxidants-10-01092-f003:**
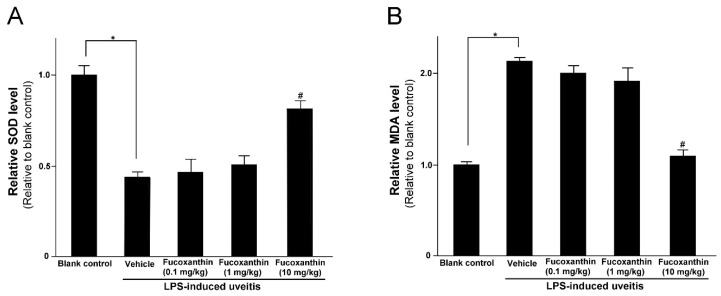
Effects of fucoxanthin on the SOD activity and MDA levels in ocular tissues treated with LPS. Treatment with 10 mg/kg BW of fucoxanthin significantly increased SOD (**A**) and decreased MDA levels (**B**), which were induced by LPS. The results represent the mean ± SD (*n* = 5) from five independent experiments. * *p* < 0.05: Significantly different compared to the blank control group (Student’s *t*-test). # *p* < 0.05: Significantly different in the LPS-treated groups (one-way ANOVA followed by Bonferroni’s multiple comparison test).

**Figure 4 antioxidants-10-01092-f004:**
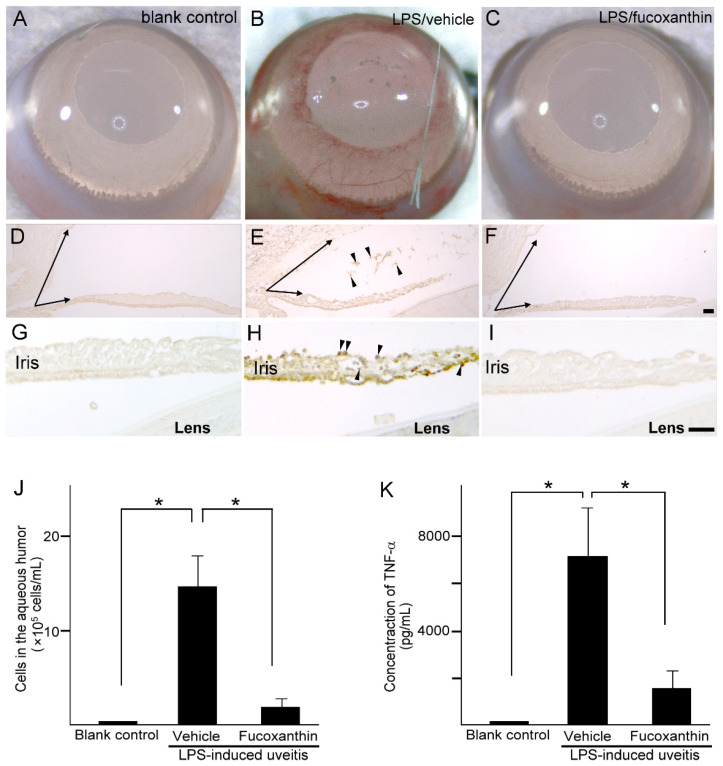
Effects of fucoxanthin on iris hyperemia, inflammatory cells, and TNF-α protein concentrations. Compared to the blank control group (**A**), iris hyperemia was observed in the LPS/vehicle group (**B**). These signs were alleviated in the LPS/10 mg/kg BW of fucoxanthin group (**C**). Moreover, the abundance of inflammatory cells was evaluated by immunohistochemical analysis of MPO-positive leukocytes in the blank control (**D**,**G**), LPS/vehicle (**E**,**H**), and LPS/fucoxanthin (**F**,**I**) groups. A narrowed anterior chamber angle and number of MPO-positive leukocytes (arrowheads) in the aqueous humor (**E**) and iris (**H**) were found in the LPS/vehicle group compared to the control group (**D**,**G**). In contrast, a relatively wider anterior chamber angle and a decreased number of MPO-positive leukocytes were observed in the aqueous humor (**F**) and iris (**I**) of the LPS/fucoxanthin group compared to that of the LPS/vehicle group. In addition, the aqueous humor was collected to count the number of cells (**J**) and to measure the expression levels of the inflammatory cytokine TNF-α (**K**). The cell count and TNF-α expression levels were increased in the LPS/vehicle group, whereas the number of inflammatory cells and the TNF-α levels in the fucoxanthin-treated groups were significantly decreased. The results are presented as the mean ± SD (*n* = 5). * *p* < 0.05: Compared to the LPS/vehicle group (Student’s *t*-test). Scale bars: 100 μm.

**Figure 5 antioxidants-10-01092-f005:**
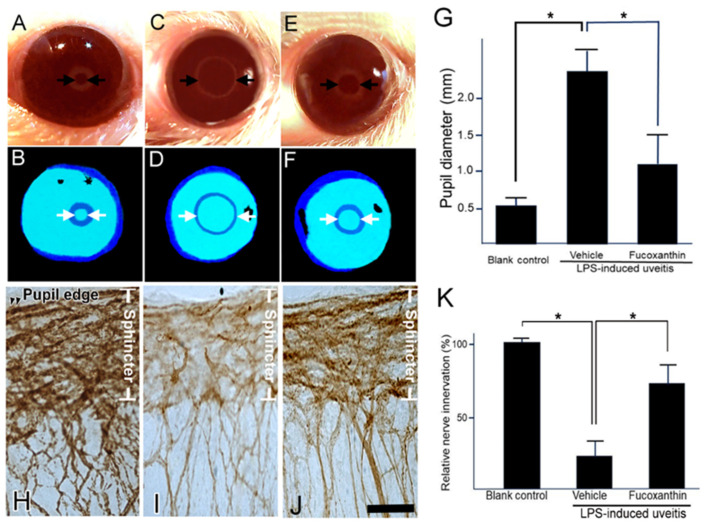
Protective effects of fucoxanthin on LPS-induced abnormal pupillary light reflex and denervation of the iris. The pupil diameter in response to pupillary light was compared among the blank control (**A**,**B**), LPS/vehicle (**C**,**D**), and LPS/10 mg/kg BW of fucoxanthin (**E**,**F**) groups. Abnormal pupillary light reflex was observed in the LPS/vehicle group relative to the control group, whereas the impaired reflex was alleviated in the LPS/10 mg/kg BW of fucoxanthin group (**G**). In addition, nerve innervation in the sphincter area was evaluated via immunohistochemical analysis of the general neural marker PGP 9.5. Nerve innervation was also compared among the blank control (**H**), LPS/vehicle (**I**), and LPS/fucoxanthin (**J**) groups. There was a significant reduction in nerve innervation in the sphincter muscle of the iris with evident LPS-induced nerve injury in the LPS-treated group (**I**) relative to in the blank control group (**H**), whereas nerve innervation was increased by treatment with fucoxanthin (**J**). Semi-quantification analysis of nerve innervation was calculated in the sphincter area and the results are presented as the mean ± SD (*n* = 5). Denervation in the sphincter region of the iris was reduced in the 10 mg/kg BW of fucoxanthin pretreatment group as compared to the LPS group (**K**). * *p* < 0.05: Compared to the LPS/vehicle group (Student’s *t*-test). Scale bars: 50 μm.

**Figure 6 antioxidants-10-01092-f006:**
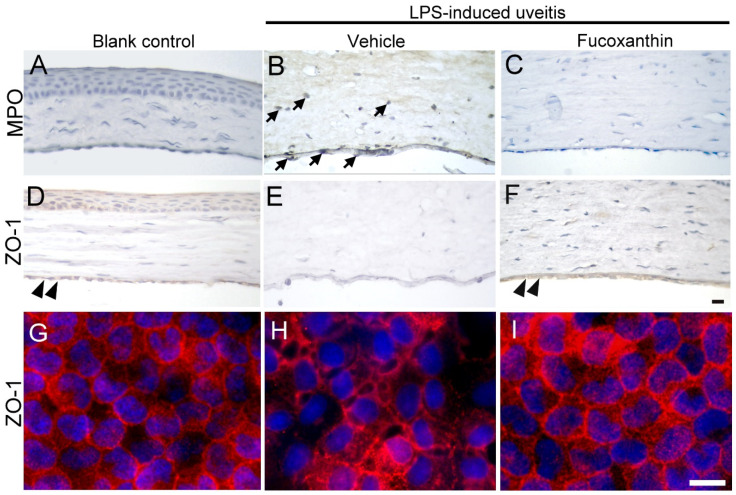
Effects of fucoxanthin on the infiltration of LPS-induced MPO-positive cells and disruption of ZO-1 in the corneal endothelium of the anterior chamber. MPO (**A**–**C**) and ZO-1 (**D**–**I**) expression levels among the blank control (**A**,**D**,**G**), LPS/vehicle (**B**,**E**,**H**), and LPS/10 mg/kg fucoxanthin (**C**,**F**,**I**) groups. Immunohistochemical staining showed strong adherent effects and infiltration of MPO-positive inflammatory cells (arrows) in the LPS/vehicle group (**B**) relative to equivalent measurements in the control group (**A**). In contrast, there were decreased numbers of MPO-positive inflammatory cells in the anterior chamber in the fucoxanthin/LPS group (**C**). Moreover, histological examination showed a continuous brown stained ZO-1 in the region of cell–cell contact in the corneal endothelium of the blank control (double arrowheads, (**D**)). Disruption of ZO-1 was observed in the LPS/vehicle group (**E**) relative to in the control group. Compared to the LPS group (**E**), an increase in the intensity of browning staining ZO-1 expression at the cell–cell junction was noted in the LPS/fucoxanthin group (double arrowheads, (**F**)). On whole flat-mounted corneas, disruption of tight junction was prevented in the LPS/10 mg/kg fucoxanthin group (**G**–**I**). Nuclei were stained with hematoxylin or DAPI. Scale bars = 20 μm.

## Data Availability

The data presented in this study are available in this paper.

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
