# Peer review of "Protective Effects of Fucoxanthin Dampen Pathogen-Associated Molecular Pattern (PAMP) Lipopolysaccharide-Induced Inflammatory Action and Elevated Intraocular Pressure by Activating Nrf2 Signaling and Generating Reactive Oxygen Species"

_antioxidants, 2021, doi:10.3390/antiox10071092_

Round 1
Reviewer 1 Report
Overview
In this well-written study, Chen et al. investigate the effects of fucoxanthin, a well-studied marine xanthophyll carotenoid antioxidant, on LPS-induced uveitis in adult male Sprague-Dawley rats. The authors report that orally administered fucoxanthin dose-dependently increased the protein expression and nuclear translocation of Nrf2 with a concomitant increase in SOD activity and decrease in MDA levels in unspecified ocular tissue harvested from treated rats. Oral fucoxanthin is also shown to protected against LPS-induced increases in IOP, prevented LPS-dependent increases in inflammatory mediators in AH, and prevented LPS-induced corneal endothelial dysfunction. Pupillary function was preserved by fucoxanthin. The authors conclude that fucoxanthin may protect against PAMP-induced uveitis by promoting Nrf2 pathway and inhibiting oxidative stress.
Specific Comments
- General: Nicely written and impressive study demonstrating clear prophylactic protection of orally administered fucoxanthin against LPS-induced uveitis. The critical challenge is, of course, whether this isoprenoid can therapeutically attenuate or reverse established on-going uveitis. The authors are encouraged to comment on this clinically relevant limitation.
- Line 88: The authors should appropriately cite previous studies investigating the effects of fucoxanthin on LPS-induced uveitis (eg, Shiratori et al. 2005; Peng et al., 2011) and clearly emphasize what is exactly novel about their study.
- Line 108: Fucoxanthin is said to have been administered orally as a 0.1 ml suspension in PBS/day x 2 days. The authors should state the method used (gavage?) and indicate how this solution was prepared. At the highest dose administered, fucoxanthin is most likely a suspension unless first dissolved in DMSO and subsequently diluted (10 mg/kg bw x 0.25 kg rat = 2.5 mg fucoxanthin in 0.1 ml = 25 mg; solubility of fucoxanthin in DMSO is ~20 mg/ml). It is unlikely that this concentration of fucoxanthin, as stated, is completed dissolved.
- Line 112: Given concerns regarding low bioavailability of fucoxanthin, the authors may wish to comment on the plasma / ocular concentration typically achieved following oral administration of this carotenoid.
- Line 142: The authors should either state the type of rotor used or should convert rpm to g forces used.
- Figure 1: While informative, relative changes in Nrf-2 protein content do not reflect functional changes. The authors should comment on this limitation, even in the presence of elevated SOD activity (Figure 3).
- Figure 1B: Does fucoxanthin promote/facilitate dissociation of Nrf2 from Keap1? The authors should comment on this mechanistic possibility.
- Figure 2: The effect of fucoxanthin on IOP is interesting. The authors should comment on a possible mechanism to explain this observation.
- Methods: It is appreciated that the authors harvested and analyzed “ocular tissue” in this study, but a paragraph is needed specifying how this tissue was prepared.
- Legends to Figures 1, 4, 5, and 6 should include the N’s used for the various data sets shown. Images should be referred to as representative.
- Legend to Figure 6: This legend is incorrectly labeled as Pure 6. Also, panels G & H are not discussed in the text (lines 321-334).
Author Response
Reviewer #1
- Line 88: The authors should appropriately cite previous studies investigating the effects of fucoxanthin on LPS-induced uveitis (eg, Shiratori et al. 2005; Peng et al., 2011) and clearly emphasize what is exactly novel about their study.
Thanks for your kindly reminder. We have added previous studies investigating the effects of fucoxanthin on LPS-induced uveitis and emphasized the exact novel about their study.
- Line 108: Fucoxanthin is said to have been administered orally as a 0.1 ml suspension in PBS/day x 2 days. The authors should state the method used (gavage?) and indicate how this solution was prepared. At the highest dose administered, fucoxanthin is most likely a suspension unless first dissolved in DMSO and subsequently diluted (10 mg/kg bw x 0.25 kg rat = 2.5 mg fucoxanthin in 0.1 ml = 25 mg; solubility of fucoxanthin in DMSO is ~20 mg/ml). It is unlikely that this concentration of fucoxanthin, as stated, is completed dissolved.
Thank you for the correction. We have corrected those in the revised manuscript.
- Line 142: The authors should either state the type of rotor used or should convert rpm to g forces used.
Thanks for your kindly reminder. We have made the changes in the revised manuscript.
- Figure 1: While informative, relative changes in Nrf-2 protein content do not reflect functional changes. The authors should comment on this limitation, even in the presence of elevated SOD activity (Figure 3).
Thank you for the constructive suggestions. We have explained and discussed it in the revised manuscript.
- Figure 1B: Does fucoxanthin promote/facilitate dissociation of Nrf2 from Keap1? The authors should comment on this mechanistic possibility.
Thank you for the constructive suggestions. The study demonstrated that fucoxanthin directly binds to Keap1 protein and increase the Nrf2-dependent antioxidant response elements (Wu et al., 2021). We have explained and discussed it in the revised manuscript.
- Figure 2: The effect of fucoxanthin on IOP is interesting. The authors should comment on a possible mechanism to explain this observation.
Thank you for the constructive suggestions. We have added the section in the revised manuscript.
- Methods: It is appreciated that the authors harvested and analyzed “ocular tissue” in this study, but a paragraph is needed specifying how this tissue was prepared.
Thank you for the constructive suggestions. We have added the section in the revised manuscript.
- Legends to Figures 1, 4, 5, and 6 should include the N’s used for the various data sets shown.
Thank you for the correction. We have added the number of the data in the revised manuscript.
- Legend to Figure 6: This legend is incorrectly labeled as Pure 6. Also, panels G & H are not discussed in the text (lines 321-334).
Thank you for the correction. We have made the changes in the manuscript.
References listed:
Wu W, Han H, Liu J, Tang M, Wu X, Cao X, Zhao T, Lu Y, Niu T, Chen J, Chen H. Fucoxanthin prevents 6-OHDA-induced neurotoxicity by targeting Keap1. Oxid Med Cell Longev. 2021:6688708.
Reviewer 2 Report
The authors elucidated the mechanism of the specific anti-inflammatory effect of fucoxanthin on pathogen-associated molecular pattern (PAMP) lipopolysaccharide-induced uveitis.
Fucoxanthin was found to effectively promote the expression of nuclear factor erythroid 2-related factor 2 (Nrf2) in ocular tissues. Furthermore, fucoxanthin significantly increased the ocular activity of superoxide dismutase and decreased the level of malondialdehyde stimulated by PAMP-induced uveitis. Levels of tumor necrosis factor-alpha, an inflammatory cell and proinflammatory cytokine in ocular hypertension, were reduced. Furthermore, PAMP-induced damage to the corneal endothelium was
significantly inhibited by the administration of fucoxanthin.
This study is well designed, the data are clean, and the paper is well written.
Minor points
#1. How about the trend of caspase3 by fucoxanthin administration?
, even if it is an excerpt from a previously reported case.
#2. How about the improvement of mitochondrial dysfunction by fucoxanthin administration?
Please describe it in Discussion, even if it is an excerpt from the reported cases.
Author Response
Reviewer #2
- How about the trend of caspase3 by fucoxanthin administration?, even if it is an excerpt from a previously reported case.
Thanks for your kindly reminder. Previous results indicated that the fucoxanthin have the dual functions of a cytoprotective effect and the apoptosis induction dependent on the treated concentrations. The results suggest that fucoxanthin against oxidative damage through down-regulation of inflammasome components and inhibiting apoptosis (Park et al., 2020; Rodríguez-Luna et al., 2019). Other studies indicated that fucoxanthin treatments induce apoptosis through caspase-3 activation in various cancer cells (Terasaki et al., 2020; Zhu et al., 2018).
- How about the improvement of mitochondrial dysfunction by fucoxanthin administration? Please describe it in Discussion, even if it is an excerpt from the reported cases.
Previous study demonstrated that fucoxanthin activates the AMP-activated protein kinase signaling pathway and inhibition of mitochondria dysfunction induced by oxidative stress (Jang et al., 2018). In addition, fucoxanthin act as antioxidants to block upstream triggers ROS-associated mitochondrial dysfunction (Wu et al., 2021). We have modified this paragraph according to your suggestion.
References listed:
- Jang EJ, Kim SC, Lee JH, Lee JR, Kim IK, Baek SY, Kim YW. Fucoxanthin, the constituent of Laminaria japonica, triggers AMPK-mediated cytoprotection and autophagy in hepatocytes under oxidative stress. BMC Complement Altern Med. 2018. 8:97.
- Park HA, Hayden MM, Bannerman S, Jansen J, Crowe-White KM. Anti-apoptotic effects of carotenoids in neurodegeneration. Molecules. 2020. 25:3453.
- Rodríguez-Luna A, Ávila-Román J, Oliveira H, Motilva V, Talero E. Fucoxanthin and rosmarinic acid combination has anti-inflammatory effects through regulation of NLRP3 inflammasome in UVB-Exposed HaCaT Keratinocytes. Mar Drugs. 2019. 17:451.
- Terasaki M, Kimura R, Kubota A, Kojima H, Tanaka T, Maeda H, Miyashita K, Mutoh M. Continuity of tumor microenvironmental suppression in AOM/DSS mice by fucoxanthin may be able to track with salivary glycine. In Vivo. 2020. 34:3205-3215.
- Wu W, Han H, Liu J, Tang M, Wu X, Cao X, Zhao T, Lu Y, Niu T, Chen J, Chen H. Fucoxanthin prevents 6-OHDA-induced neurotoxicity by targeting Keap1. Oxid Med Cell Longev. 2021 :6688708.
- Zhu Y, Cheng J, Min Z, Yin T, Zhang R, Zhang W, Hu L, Cui Z, Gao C, Xu S, Zhang C, Hu X. Effects of fucoxanthin on autophagy and apoptosis in SGC-7901cells and the mechanism. J Cell Biochem. 2018. 119:7274-7284.
Reviewer 3 Report
In the LPS-induced uveitis model in rats, fucoxanthin, one of carotenoids, increased SOD levels, inhibited malondialdehyde production, and suppressed TNF-alpha elevation and inflammation by increasing the expression of Nrf2, a factor involved in antioxidation.
In this study, the anti-inflammatory effect of fucoxanthin has been well demonstrated. It is expected that the application of fucoxanthin will be expanded. However, there seem to be some problems in this study, such as some unclear points in the experimental method, unsuitable statistical processing, and unfortunately, fucoxanthin is hardly absorbed in the intestinal tract in human, so the test in rats cannot be adapted to humans.
1, Fucoxanthin is a fat-soluble component. How did the authors dissolve it in PBS? 
2, Regarding the method of administering fucoxanthin to rats, is it forced administration by a sonde? The detailed method is not described in this manuscript.
3, For the statistical methods in all the figures, it appears that the Dunnett test described in the methods section (line 190) is not actually used. Also, the T-test cannot be used to test for multiple groups. The authors compare two of the five groups, or two of the three groups, with the T test, which should not be done. Since the authors are doing a test for multiple groups (Bonferroni's multiple comparison test), that should be used to compare between all the individual groups.
4, Since there is no comparison between fucoxantihn and the positive control, it is not shown how much effect fucoxanthin has. It seems that the authors need to show this. Also, it should be discussed whether the anti-inflammatory effect is the antioxidant effect of fucoxanthin or that of carotenoids. It has been reported that lutein and zeaxanthin, which accumulate specifically in the retina, enhance the expression of Nrf2. For example, the following.
J Cell Commun Signal. 2020 Jun;14(2):207-221.
J Agric Food Chem. 2017 Jul 26;65(29):5944-5952.
Biochem Biophys Res Commun. 2019 Aug 13;516(1):163-170.
In other words, the anti-inflammatory effects may be possessed by carotenoids and not limited to fucoxanthin.
5, Fucoxanthin is hardly absorbed from the intestinal tract in humans. At least, it can be said that fucoxanthin is not absorbed at all in humans compared with beta-carotene and lutein (Low bioavailability of dietary epoxyxanthophylls in humans. Br J Nutr. 2008;100(2):273-7.)
On the other hand, it is known that fucoxanthin is also absorbed from the intestinal tract in rats and mice, unlike humans. In this study, in rats, is fucoxanthin transported to the eye through the blood-eye barrier after intestinal absorption to exert its antioxidant (anti-inflammatory) effect? Or, does fucoxanthin simply pass through the intestinal tract to activate Nrf2? The authors should discuss these points.
Even if the medicinal ingredients are not absorbed from the intestinal tract, there is a possibility that they can reach the eye from eye drops. In this study, if the authors are considering the application of fucoxanthin to human eyes, they should have examined the effect of fucoxanthin by eye drops instead of oral.
6, Minor mistakes. For example,
Line 336, Pure 6 → Figure 6
Author Response
Reviewer #3
- Fucoxanthin is a fat-soluble component. How did the authors dissolve it in PBS?
Thanks for your kindly reminder. Fucoxanthin at 0.1, 1, or 10 mg/kg BW in a 0.1% dimethyl sulfoxide solution mixed with 0.1 mL PBS. We have made the changes in the revised manuscript.
- Regarding the method of administering fucoxanthin to rats, is it forced administration by a sonde? The detailed method is not described in this manuscript.
Thank you for the constructive suggestions. We have described in the revised manuscript.
- For the statistical methods in all the figures, it appears that the Dunnett test described in the methods section (line 190) is not actually used. Also, the T-test cannot be used to test for multiple groups. The authors compare two of the five groups, or two of the three groups, with the T test, which should not be done. Since the authors are doing a test for multiple groups (Bonferroni's multiple comparison test), that should be used to compare between all the individual groups.
Thanks for your kindly reminder. Differences between two groups were analyzed using a student t test and the mean value was significantly different as compared with the blank control group. Significantly differenced in LPS-induced group were using the one-way ANOVA followed by Bonferroni’s multiple comparison. In order to be easily to read, we have the replaced the figures 2 and 3 pictures and made changes in the revised manuscript.
- Since there is no comparison between fucoxantihn and the positive control, it is not shown how much effect fucoxanthin has. It seems that the authors need to show this. Also, it should be discussed whether the anti-inflammatory effect is the antioxidant effect of fucoxanthin or that of carotenoids. It has been reported that lutein and zeaxanthin, which accumulate specifically in the retina, enhance the expression of Nrf2. For example, the following.
J Cell Commun Signal. 2020 Jun;14(2):207-221.
J Agric Food Chem. 2017 Jul 26;65(29):5944-5952.
Biochem Biophys Res Commun. 2019 Aug 13;516(1):163-170.
In other words, the anti-inflammatory effects may be possessed by carotenoids and not limited to fucoxanthin.
Thank you for the constructive suggestions. Although it is well known that the retina is enriched with the carotenoid lutein, relatively little is known regarding fucoxanthin levels in ocular tissues. Previous study indicated fucoxanthin suppresses the endotoxin-induced inflammation in a reduction in PGE2, NO, and TNF-alpha concentrations in the aqueous humour (Shiratori et al., 2005). Because the cornea is composed of avascular tissue, nutrition of the corneal tissues must occur via diffusion of solutes from the aqueous humor. In the study, we demonstrated that the treatment with fucoxanthin may protect against LPS-induced disorders by inhibiting expression of proinflammatory factors.
- Fucoxanthin is hardly absorbed from the intestinal tract in humans. At least, it can be said that fucoxanthin is not absorbed at all in humans compared with beta-carotene and lutein (Low bioavailability of dietary epoxyxanthophylls in humans. Br J Nutr. 2008;100(2):273-7.)
On the other hand, it is known that fucoxanthin is also absorbed from the intestinal tract in rats and mice, unlike humans. In this study, in rats, is fucoxanthin transported to the eye through the blood-eye barrier after intestinal absorption to exert its antioxidant (anti-inflammatory) effect? Or, does fucoxanthin simply pass through the intestinal tract to activate Nrf2? The authors should discuss these points. Even if the medicinal ingredients are not absorbed from the intestinal tract, there is a possibility that they can reach the eye from eye drops. In this study, if the authors are considering the application of fucoxanthin to human eyes, they should have examined the effect of fucoxanthin by eye drops instead of oral.
Thank you for the constructive suggestions. In the study, we demonstrated that the treatment with fucoxanthin may protect against LPS-induced disorders by inhibiting expression of proinflammatory factors in corneal tissues. In addition, the study demonstrated that fucoxanthin could directly binds to Keap1 protein and increase the Nrf2-dependent antioxidant response elements (Wu et al., 2021). We have discussed it in the revised manuscript.
- Line 336, Pure 6 → Figure 6
Thank you for the correction. We have corrected those in the revised manuscript.
References listed:
- Shiratori K, Ohgami K, Ilieva I, Jin XH, Koyama Y, Miyashita K, Yoshida K, Kase S, Ohno S. Effects of fucoxanthin on lipopolysaccharide-induced inflammation in vitro and in vivo. Exp Eye Res. 2005, 81: 422-8.
- Wu W, Han H, Liu J, Tang M, Wu X, Cao X, Zhao T, Lu Y, Niu T, Chen J, Chen H. Fucoxanthin prevents 6-OHDA-induced neurotoxicity by targeting Keap1. Oxid Med Cell Longev. 2021:6688708.